# On the High Structural Heterogeneity of Fe-Impregnated Graphitic-Carbon Catalysts from Fe Nitrate Precursor

**Rosa Arrigo** [1,2,*] and **Manfred Erwin Schuster** [3,*]

1   School of Environment & Life Sciences, University of Salford, Cockcroft building, Greater Manchester M5 4WT, UK
2   Diamond Light Source Ltd., Harwell Science & Innovation Campus, Didcot, Oxfordshire OX11 0DE, UK
3   Johnson Matthey Technology Centre, Blount's Court Road, Sonning Common, Reading RG4 9NH, UK
*   Correspondence: r.arrigo@salford.ac.uk or rosa.arrigo@diamond.ac.uk (R.A.); manfred.schuster@matthey.com (M.E.S.)

**Abstract:** Wet impregnation is broadly applied for the synthesis of carbon-supported metal/metal oxide nanostructures because of its high flexibility, simplicity and low cost. By contrast, impregnated catalysts are typified by a usually undesired nanostructural and morphological heterogeneity of the supported phase resulting from a poor stabilization at the support surface. This study on graphite-supported Fe-based materials from Fe nitrate precursor is concerned with the understanding of the chemistry that dictates during the multistep synthesis, which is key to designing structurally homogeneous catalysts. By means of core-level X-ray photoelectron spectroscopy, near-edge X-ray absorption fine structure spectroscopy and atomic resolution electron microscopy, we found not only a large variety of particles sizes and morphologies but also chemical phases. Herein, thermally stable single atoms and few atoms clusters are identified together with large agglomerates of an oxy-hydroxide ferrihydrite-like phase. Moreover, the thermally induced phase transformation of the initially poorly ordered oxy-hydroxide phase into several oxide phases is revealed, together with the existence of thermally stable N impurities retained in the structure as Fe–N–O bonds. The nature of the interactions with the support and the structural dynamics induced by the thermal treatment rationalize the high heterogeneity observed in these catalysts.

**Keywords:** ferrihydrite; Fe2p; O1s; N1s XPS; NEXAFS; HAADF-STEM

## 1. Introduction

Fe oxides and oxyhydroxides exist in nature in a large variety of structures and play a primary role in many natural geological and biological processes [1]. Their reactivity is also exploited in many synthetic chemical processes. For instance, Fe/C systems are widely investigated as alternatives to platinum group metals systems in electro-catalytic applications such as the oxygen reduction reaction [2–5], the electrochemical $CO_2$ reduction [6,7], the electrochemical $NH_3$ synthesis [8] as well as in thermo-catalytic applications such as the Fischer–Tropsch synthesis [9–11]. Another application of Fe/C systems is as a sorbent for the removal of heavy metals from aqueous systems [12]. A common preparation route for carbon-supported Fe oxide and oxyhydroxide nanoparticles (NPs) is wet impregnation which consists of three steps: a) the wetting of the solid support with an aqueous solution containing a precursor of the active metal, e.g., ferric or ferrous nitrate, chloride and acetate; b) the drying at a given temperature and normally in air; and c) the thermal treatment in an inert or reactive atmosphere to decompose the precursor and produce the desired phase [13]. The popularity of this preparation route is attributable to its simple execution and the use of inexpensive

laboratory equipment. In an ideal scenario, the synthesis protocol is designed in such a way that, during the impregnation step, electrostatic interactions lead the molecules of the metal precursor to diffuse into the solution and to approach the hydrophilic region of the carbon surface where homogeneously distributed functional groups of opposite charge are located at the edges or vacancies of the graphitic structure. The formed liquid/solid interface behaves as an electrical double layer as, for instance, the one described by the Gouy–Chapman–Stern model [14]. There, the functional groups on the support can also establish a chemical bond with the complexes in solution, namely, specific chemisorption. A homogeneous chemical composition at the solution/support interface is, therefore, a necessary prerequisite to attain monodispersed NPs. In reality, the acid-base properties of the functional groups at the carbon surface and the carbon structure itself are highly heterogeneous in nature [15,16]. Also, the chemistry of the ferric species in solution is characterized by a complex network of chemical equilibria strongly influenced by the local pH: a) for a ferric solution at an approx. pH 2, mono-hydroxylated and bi-hydroxylated complexes, $(Fe(III)(OH)(OH_2)_5)^{2+}$ and $(Fe(III)(OH)_2(OH_2)_4)^+$, were found to be the most abundant species in the bulk of the solution [17] (whilst with increasing pH, the charge on the ion complex decreases); b) in the absence of a complexing agent, these ferric species in solution condense very rapidly at pH > 3 (through oletion or oxolation reactions [17]) to form polycationic species of different nuclearity. Thus, the Fe component on the carbon surface is expected to already vary significantly in charge and nuclearity in the sample precursor state. Furthermore, during the drying process, the gradual supersaturation of the bulk of the solution with the dissolved species could lead to the formation of poorly crystalline precipitates depositing onto the carbon support with no specific interactions with it. Finally, during the thermal annealing, the ferric phase undergoes chemical reduction, and particles not strongly interacting with the carbon surface tend to agglomerate leading to a poor dispersion. The phase obtained depends on the temperature and atmosphere of the thermal treatment, but an influence of the initial Fe phase present after drying is expected. It must be pointed out that the subsequent exposure to air will reoxidize the Fe phase [6] leading to nanoparticles with a variety of chemical phases and configurations (i.e., nanoparticles with an Fe oxide shell on a metallic Fe core were found in Reference [7]). For the reasons listed above, the control of the chemical phase, dispersion and particle morphology remains a challenge of this synthetic route. In view of the popularity that carbon supports have gained in the field of electro-catalysis and the high accessibility of this synthetic route, we provide here a comprehensive structural characterization of C-supported Fe-based catalysts obtained via wet impregnation of a ferrous nitrate solution onto graphitic C felt, with the aim to contribute towards a deeper understanding of the critical factors determining the catalysts' nanostructures.

Similarly to a previous work [18–20] on C nanotubes-supported Pd nanoparticles, we investigate herein carbon felt which was functionalized by using either $HNO_3$ (OC) or $NH_3$ (NC) to introduce mainly carboxylic O and pyridine N functionalities, respectively. We apply synchrotron-based X-ray photoelectron spectroscopy (XPS) and near-edge absorption fine structure (NEXAFS) spectroscopy in combination with atom resolving transmission electron microscopy (TEM). These techniques provide information about the metal oxidation states and coordination geometry as well as the chemical bonding configuration and, thus, represent most striking tools to investigate the local nanostructures of materials. The structural transformation of the Fe phase in situ upon thermal annealing will also be shown, which firstly leads to the rearrangement of the initial oxy-hydroxide phase into a more ordered ferrihydrite phase mixed together to wüstite, whilst at a higher temperature, magnetite is also formed. Additionally, this work has led to the identification of a phase containing Fe–N species with an unexpected high thermal stability. Overall, the information obtained in this work contributes a rationale towards a controlled synthesis of C-supported materials.

## 2. Results

Functionalized carbon felts (OC and NC) were impregnated with a Fe nitrate solution to synthetize Fe oxide structures after an annealing step in $N_2$ at 473 K and exposure to an atmospheric environment.

The ferric solution with a measured pH of approximately 2.3 was quantitatively loaded onto the carbon support with the targeted total loading of 1 percent in weight expressed as metallic Fe. Hereafter, the samples will be referred to as Fe/OC and Fe/NC. An additional sample was prepared via an ionic exchange from an iron chloride aqueous solution (referred to as $Fe_{(I.E.)}$/NC) as a reference for the spectroscopic analysis of Fe, N and O species in the impregnated samples.

*2.1. Electron Microscopy Structural Analysis*

Representative scanning electron micrographs (SEM) of the Fe/OC sample are reported in Figure 1a–d in secondary electrons (SE) and back-scattered electrons (BSE) modes. The same morphologies were found on the Fe/NC and already reported in Reference [6]. Accordingly, the carbon felt is made of entangled fibers of high graphitic character as seen in the high G band (ordered carbon) to D band (disordered carbon) ratio [21,22] in the Raman spectrum of the Fe/O–C samples shown in Figure 2. We also notice the presence of thin layers in the interstitial space between entangled fibers, which also show, in the Raman spectrum, a predominant G-band of graphite (not included). The Fe phase forms on the carbon support as a porous overlayer as seen in the bright area of the BSE image (Figure 1b,d). Moreover, bigger particles appear physically entrapped in the interstitial spaces between entangled fibers. These Fe nanostructures are irregularly shaped but composed of agglomerated smaller crystallites as depicted in the transmission electron micrograph (TEM) (Figure 3a). The corresponding fast Fourier transform (FFT) shows typical reflections of the graphitic support and, for some more crystalline particles, the ferrihydrite phase (Figure 3b), an *hcp* Fe(III) oxyhydroxide in which Fe is found in both octahedral and tetrahedral coordination geometries bound to both O bridge and terminal OH ligands [23]. The TEM analysis of this work is consistent with the bulk structural characterization of these materials, which was assessed in a previous publication [6] by means of Fe K edge X-ray absorption spectroscopy. The majority of the Fe nanostructures assume the same predominant phase and morphologies on both O and N functionalized supports, which suggests a mechanism of formation involving the initial generation of clusters in solution (probably Fe13 "Keggin" clusters [24]), followed by their aggregation to form nanoparticles. Upon the thermal annealing performed in the final step of the preparation, the smaller crystallites randomly deposited on the roughened carbon surface condense further into a more extended oxy-hydroxide layer [6]. Figure 4a–c reports the bright field (BF), the high-angle annular dark field (HAADF) and the SE scanning transmission electron microscopy (STEM) micrographs of Fe/OC, respectively. Note that the HAADF mode is sensitive to the atomic number, and therefore, heavier elements appear brighter, whereas the SE provides information of the surface morphology.

By comparing the micrographs in the three different modes it is possible to identify the location of small clusters and single atoms embedded in a carbon matrix with carbon overlayers decorating the clusters. These findings also suggest that single atoms and clusters are decorating the edges and vacancies of the graphitic layers and are formed in the first step of the synthesis through complexation by the functional groups on the support surface. It is evident here that, due to the low availability of chemisorption sites (edges and vacancies) on the graphitic felt used in this work, the amount of highly dispersed single sites and clusters is very low, whereas the majority of the Fe nanostructures are formed by the unspecific adsorption and deposition of clusters and NPs formed in solution and upon annealing onto the graphitic support.

*2.2. X-ray Photoemission Spectroscopy and Absorption Spectroscopy in Ultra High Vacuum (UHV)*

The surface and near surface quantitative elemental composition determined by XPS is reported in Table 1.

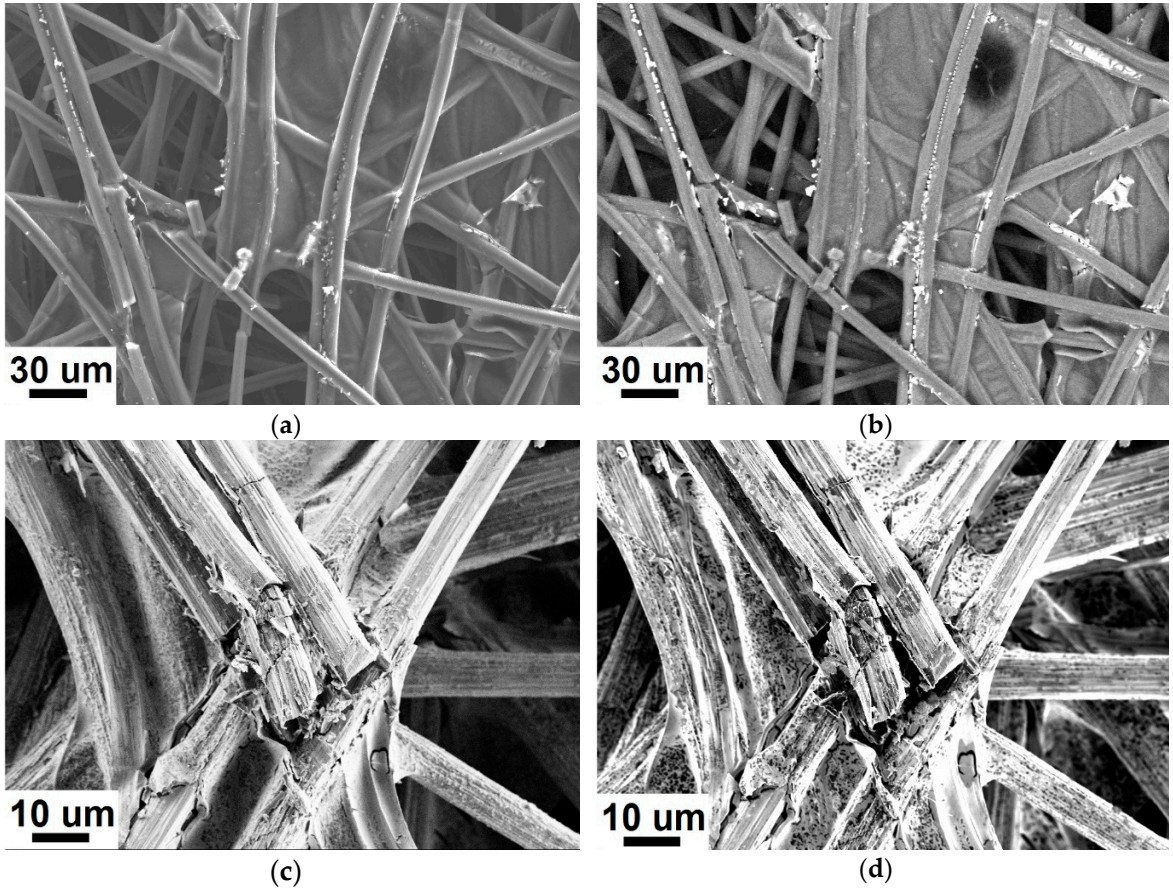

**Figure 1.** Scanning electron micrographs for the sample Fe/OC: (**a**) A low-magnification secondary electron (SE) mode image; (**b**) a low-magnification back-scattered electron (BSE) mode image; (**c**) a high-magnification SE mode image; and (**d**) a high-magnification BSE mode image.

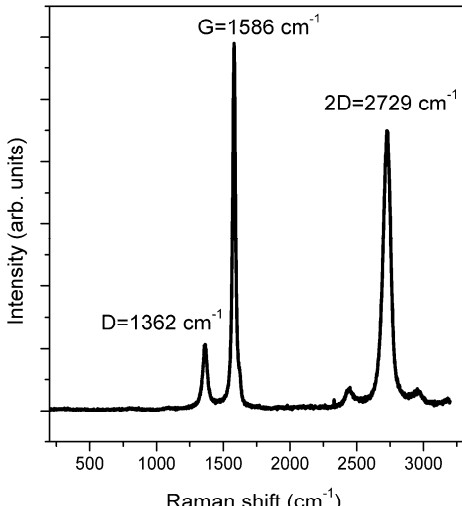

**Figure 2.** The Raman spectrum of the sample Fe/OC: The D band at 1362 $cm^{-1}$ is assigned to defective carbon whereas the G band at 1586 $cm^{-1}$ and the 2D band at 2729 $cm^{-1}$ are assigned to ordered graphite.

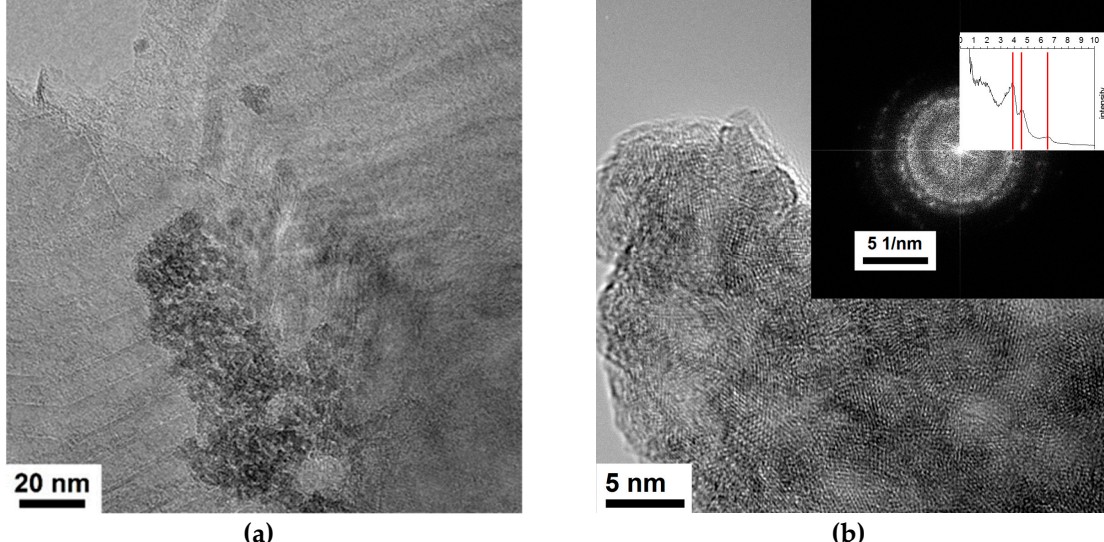

**Figure 3.** (**a**) A transmission electron micrograph of Fe/NC and (**b**) a high-resolution transmission electron micrograph of an agglomerate of Fe-rich nanoparticles in Fe/NC and the corresponding fast Fourier transform (FFT) pattern (inset): The reflexes marked in red in the radial distribution function correspond to 2.6A and 1.5A and are attributed to ferrihydrite, whereas the double peak in between corresponding to 2.217 Å and 2.027 Å are attributed to graphite ((010) and (011) respectively).

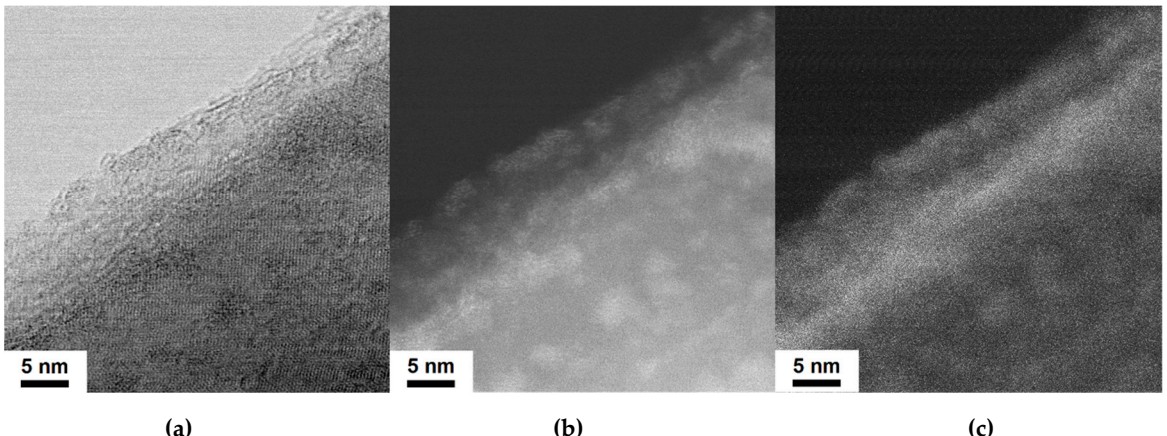

**Figure 4.** (**a**) The bright field (BF), (**b**) high-angle annular dark field (HAADF) and (**c**) SE scanning transmission electron microscopy (STEM) micrograph of Fe/NC: the brighter features in Figure 4b indicate the location of Fe on the NC support.

**Table 1.** The surface elemental composition as determined by XPS [a].

| Sample | O | N | Fe | C | Si |
|---|---|---|---|---|---|
| Fe/NC | 9.3 | 0.5 | 2.4 | 87.3 | 0.5 |
| Fe/OC | 13.6 | 0.4 | 3.6 | 81.9 | 0.5 |
| Fe$_{(I.E.)}$/NC [b] | 8.5 | 0.44 | 0.06 | 86.5 | 4.5 |

[a] The spectra were recorded by collecting photoelectrons with a kinetic energy (KE) of 450 eV, corresponding to a sampling depth of 1.5–2 nm [25]. [b] The acronym I.E. stands for ionic exchange synthetic route. Note that this sample contains a large amount of Si and that most of the oxygen is related to the Si rather than the Fe.

Noteworthy in the elemental analysis, N species are present in both the Fe/NC and Fe/OC samples obtained from a nitrate precursor as residual impurities. We compare these data with the elemental composition of a sample obtained via ionic exchange from an aqueous solution containing iron chloride (Fe$_{(I.E.)}$/NC). In this way, the amount of N species as well as their chemical speciation

is for this sample only related to the functionalization treatment of the carbon support. Accordingly, this sample presents a much lower metal loading commensurate with the ionic exchange capacity of the support, which is defined by the amount of N functional groups available for chemisorption on the C surface.

We will discuss in more detail the chemical bonding and electronic configurations of the elements by analyzing the high-resolution core levels XPS.

Figure 5a–d shows the C1s, Fe2p, O1s and N1s XP spectra measured in UHV for both the Fe/OC and Fe/NC samples by collecting electrons with a kinetic energy (KE) of 450 eV corresponding to a sampling depth of 1–2 nm [25]. The C1s spectrum presents an asymmetric peak with a maximum binding energy (BE) at 284.3 eV, which confirms the graphitic nature of the carbon support [26]. Additionally, a component at 285 eV assigned to defective C related to the N species is present in the spectrum of N-containing carbon support [15,26]. Interestingly, carboxylic oxygen moieties introduced by the treatment with $HNO_3$ [15] (expected at 288 eV circa) are not visible in the impregnated Fe/OC sample suggesting (a) that these species are not thermally stable at temperatures above 523 K and, therefore, they decompose during the thermal annealing in $N_2$ and (b) that the carboxylates species are involved in a chemical interaction with the Fe species and, due to the thickness of the nanoparticles, are not visible by the surface sensitive mode used in these XPS measurements.

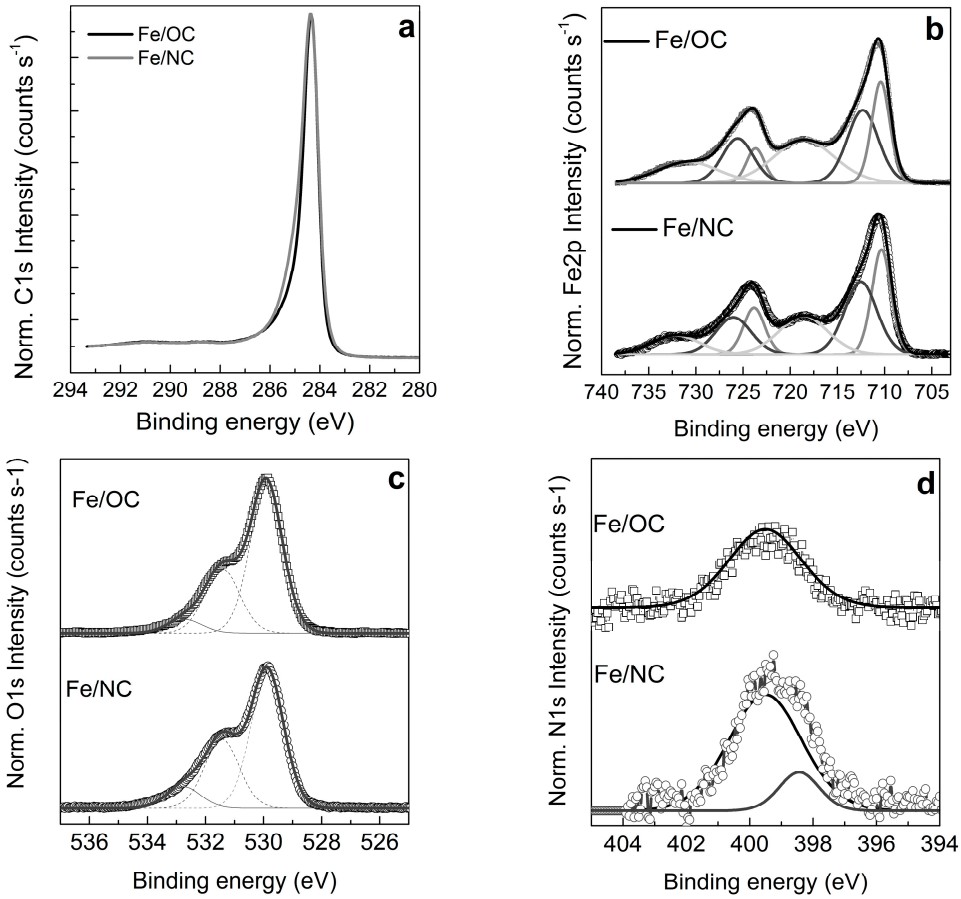

**Figure 5.** The XPS characterization of the impregnated samples Fe/OC and Fe/NC in UHV obtained by detecting electrons with a kinetic energy (KE) of 450 eV: The C1s XP spectra (**a**); Fe2p XP spectra (**b**); O1s XP spectra (**c**); and N1s XP spectra (**d**).

The analysis of the spin-orbit split Fe2p XP spectrum of iron oxides is complicated by the multiplet splitting due to the interactions of the core hole created in the photoemission process and the valence 3d electrons. Indeed, for hematite and Fe–OOH phases, a broad $Fe2p_{3/2}$ appears between 709 eV and 712 eV. This was previously fitted with 4 components describing the multiplet splitting of Fe(III)

bound to $O^{2-}$ ligands [27,28]. Additionally, the XPS peak line shape and the satellite structures are affected by both the changes in the metal Fe3d to ligands O2p hybridization parameters and the d−d electron correlation energy [29]. Here, we adopt a simplified fitting procedure that enables us to describe the differences observed in the spectra reported in this study, which represent differences in the Fe electronic structures and chemical bonding configurations. Accordingly, two components at 710.4 eV and 712.3 eV and the satellite feature at 718.4 eV were considered, which are characteristic of Fe(III) species. The second component at 712.3 eV was attributed to Fe(III) bound to $OH^-$ ligands in hydroxides [30]. The ΔBE of the spin orbit splitting is 13.5 eV.

Consistently, the corresponding O1s XP spectra in Figure 5c present a component at 530.1 eV BE assigned to $O^{2-}$ species as bridge-oxygen bound to Fe(III) and an additional component at 531.5 eV assigned to terminal OH species. The component at 533 eV is attributed to chemisorbed water, although C–O species are also found in this region. In fact, the O1s and Fe2p XP spectra resemble the spectrum reported for the Fe–OOH structures [28]. The difference spectrum indicates that the $O^{2-}$ component is more intense for the Fe/OC whereas the $OH^-$ component is more intense for the Fe/NC, suggesting more surface-exposed hydroxyls on the Fe/NC and, thus, smaller crystallites.

Moreover, the Fe2p difference spectrum indicates an additional component at 708.8 eV that is present on the Fe/NC in a very tiny amount (Figure 6a). A similar BE was found for Fe(II) species bound to ligands less electronegative than $O^{2-}$ such as in $FeBr_2$ or $K_4Fe(CN)_6$ [28]. The analysis of the N 1s core level spectra in Figure 5d was complicated by the fact that some of the N from the nitrate precursor is retained in the structure also after the thermal annealing (shown later on). This is an important aspect of the synthesis of Fe oxides from nitrate precursor via impregnation, which the presence of is normally disregarded.

For a comparison, we report the Fe2p, O1s and N1s XP spectra of $Fe_{(I.E.)}$/NC obtained from an iron chloride precursor in Figure 6b–d. The Fe2p XP spectrum shows a peak much less intense than the impregnated catalyst but broader towards the lower BE side. The two main components are at 709.4 eV and 710.7 eV, indicating a mixture of Fe(II)/Fe(III) species [28]. The O1s XP spectrum for this sample does not show the component at 530.1 eV due to $O^{2-}$ ligands, but the main peak is centered at 531.7 eV. This would suggest that Fe is present as a hydroxide or in a minor amount as chloride (not shown), although given the large amount of Si introduced by this preparation route, we expect that most of the O is indeed due to some Si oxide phase [31]. The comparison between the N1s XPS of the Fe/NC and the $Fe_{(I.E.)}$/NC allows the clear identification of the N species of the carbon support and distinguishes them from those originating from the nitrate precursor. We have previously assigned the component at 398.4 eV to pyridine species [15], whereas the BE is shifted to a higher value for coordinated pyridine [18,20,32]. As expected, the pyridine species is present in both Fe/NC and $Fe_{(I.E.)}$/NC but not on the Fe/OC. Here, we also observe that the component at 399.4 eV is due to both the N species on the supports that are coordinated to the Fe species as well as the Fe–N species found in the samples and originating from the nitrate precursor. The spectrum of the impregnated sample presents additional contributions at a higher BE (maximum at 400.4 eV in the spectrum in red in Figure 6d) assigned to Fe–N–O species. This result suggests that Fe hydroxo-nitrato complexes are formed in solution similarly to the case of Pd nitrate in concentrated nitric acid solution [33].

We now analyze the NEXAFS spectra both in the total electron yield (TEY) and the Auger electron yield (AEY) mode as specified case-by-case. Generally, TEY is thought to be more bulk sensitive than AEY; however, we find that both spectra are very similar, but the TEY enables a much higher signal to noise ratio. In the case of N, we use the AEY signal because the TEY signal in the N1s region is affected by the significant X-ray absorption at the silicon nitride window separating the analysis chamber from the beamline. The Fe2p NEXAFS spectra of the impregnated samples were already published in Reference [6] and are here reported in Figure 7a. The spectra of both samples are characterized by the resonances R1 (2p→ 3t2g) at 709 eV and R2 (2p→ 3eg) at 710.4 eV, which are typical of Fe(III) species. The difference spectrum shows a resonance at a low excitation energy for the N functionalized support which is signature of a few Fe(II) sites [6] consistently with the additional component in the

Fe2p XP spectrum in Figure 6a. Compared to hematite, the signature of the ferrihydrite structure is the Fe(III) species in tetrahedral sites which were identified at the Fe2p NEXAFS spectrum as an additional resonance of eg-t2g type between the t2g and the eg resonances of the dominant Fe(III) in Oh symmetry [34].

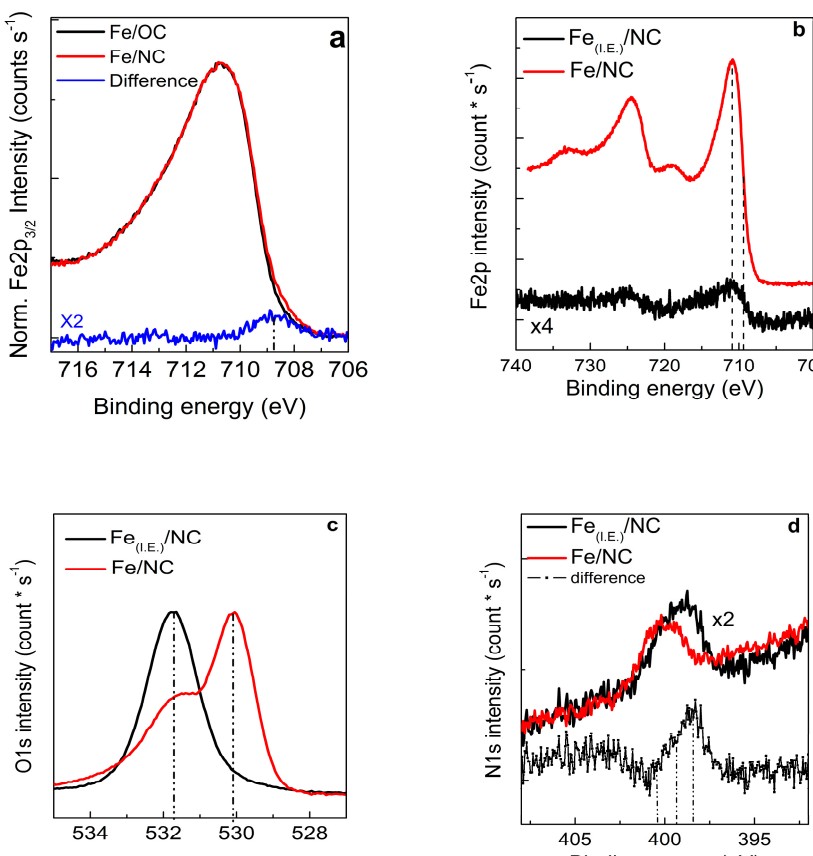

**Figure 6.** The XPS characterization in UHV obtained by detecting electrons with a kinetic energy (KE) of 450 eV: The Fe 2p XPS spectra for Fe/NC and Fe/OC and the difference spectra (**a**); the Fe2p spectra for Fe/NC and Fe$_{(I.E.)}$/NC (**b**); the O1s spectra for Fe/NC and Fe$_{(I.E.)}$/NC (**c**); and the N1s spectra for Fe/NC and Fe$_{(I.E.)}$/NC and difference (**d**).

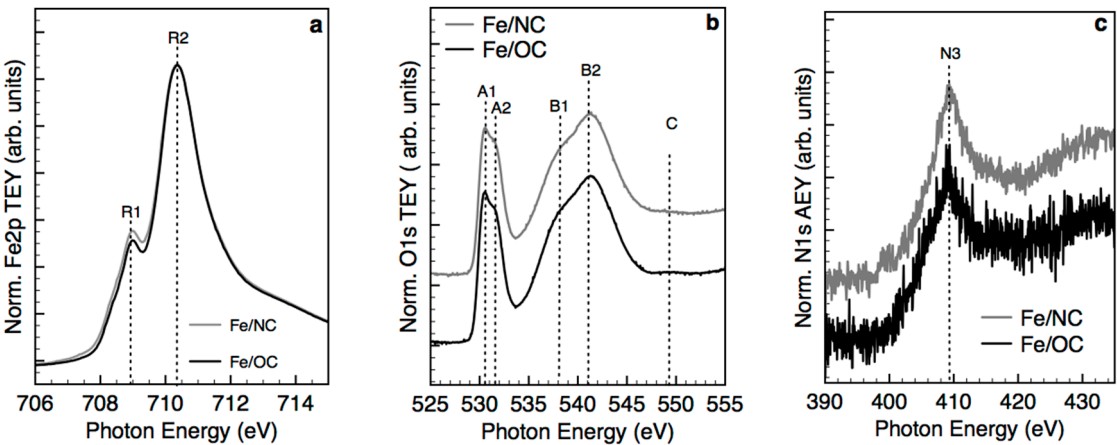

**Figure 7.** The near-edge absorption fine structure (NEXAFS) characterization of the impregnated samples Fe/OC and Fe/NC in UHV: (**a**) the Fe2p total electron yield (TEY) spectra; (**b**) the O1s TEY spectra; and (**c**) the N1s Auger electron yield (AEY) spectra.

We also analyze the O1s NEXAFS spectra for these samples. These provide information about not only the local structural geometry but also the long-range order through the scattering of the photo-emitted electrons with the absorbing atom appearing as resonances above the threshold edge.

The O1s NEXAFS spectrum of the impregnated Fe–OOH samples in Figure 7b is discussed against assignments from previous literature [35–37]: a pre-peak region composed of two resonances A1 and A2 at 530.5 eV and 531.6 eV, respectively; a broad feature composed of two main resonances B1 and B2 at 541 eV; and an additional broad resonance C above 545 eV. The split A1 and A2 resonances are assigned to transitions to O2p empty states hybridized with 3d metal states that assume t2g-eg symmetry due to the crystal field effect [35]. Differently to the spectrum reported for hematite, the doublet here is not very well-resolved whilst the energy splitting is circa 1.1 eV, which is much smaller than in the case of hematite (1.45 eV [35]). In fact, the pre-peak region of these spectra is similar to the spectra reported for maghemite and magnetite [35]. In the case of magnetite, it was suggested that the presence of mixed tetrahedral and octahedral sites, each one associated with the eg-t2g splitting, leads to a broadening of the features. This explanation could be valid for the ferrihydrite as well. Additionally, the O1s spectrum in Figure 5c has shown clearly a component at a higher BE assigned to OH species; therefore, we expect that the resonances in the O1s region originate from the transition from two chemically inequivalent oxygen species, where each peak is composed of at least two resonances.

The high-energy region of the spectrum above the edge threshold is characterized by two broad resonances B1 and B2 due to the contributions of the Fe 4sp to the O 2p density of states, whilst the extensive spread indicates the covalent character of the bonds in these materials. Whilst its intensity shows no strong dependence with the crystal structure [36], their energy separation is related to the regularity of the Fe_O6 octahedra. The ferrihydrite structure in these samples is characterized by an energy separation of 3.3 eV, which is very close to the magnetite (regular octahedron) and much bigger than the value for hematite (some distorted octahedron). These split resonances were discussed in the literature [35,37] to be originated from the multiple scattering of the final state photoelectron emitted by the oxygen absorber atom with Fe and O atoms within a radius of 3–4 Å from the central absorbing atom. The feature C was theoretically reproduced in hematite [37] by using a single oxygen shell located above 5 Å from the absorber, proving that this is predominantly due to a single scattering event between the photo-absorber and the third oxygen shell. However, the resonance C between 450–545 eV in the ferrihydrite samples is not as intense as for magnetite and hematite, indicating that the long-range order is limited in this sample. The electronic structure analysis of the surface is also consistent with the electron microscopy study, in which most of the nanoparticles are composed of agglomerated clusters with poor crystalline order (Figure 3). In addition, a few clusters and atomically dispersed species are correlated with the appearance of C reduced Fe species that are more abundant on NC, as evidenced by XPS and NEXAFS.

The N1s NEXAFS spectra of the samples obtained via wet impregnation from nitrate precursor (Figure 7c) show a peculiar shape, which differs from the sharp resonances found for Fe nitrate [38]. Particularly, the spectra are dominated by a broad 1s→σ* resonance (herein, we refer to this as N3) with a maximum at 409.6 eV, whilst the 1s→π* resonance is apparently absent. A similar spectrum was observed for Fe(II)-cysteine [38] and metal organic framework with amino ligands [39]. We consider that the N1s NEXAFS spectrum of the impregnated samples is dominated by the signal of Fe–N species originating from the Fe precursor, whereas the signal of the Fe–N species due to the interaction with the N functionalized support is overshadowed.

In order to investigate further the chemical state of N in the impregnated catalysts, we now compare them with the spectrum of the sample synthetized by ionic exchange (Fe$_{(I.E.)}$/NC), starting from a FeCl$_3$ precursor (Figure 8a–c). Of interest, the AEY Fe2p NEXAFS spectrum of this sample presents a higher intensity at a lower excitation energy, which appears in the difference spectrum as a component centered at 709.4 eV. This resonance is characteristic of Fe(II) species, and from the results obtained in this study, this resonance could likely be a highly dispersed species interacting with the carbon support. The O1s NEXAFS spectra for the Fe$_{(I.E.)}$/NC significantly differ from the impregnated

sample but resembles the one reported for silicalite [40]. More importantly, we can discriminate the resonances due to the N–C surface from the Fe–N species by comparing the N1s NEXAFS spectra of the two samples obtained via impregnation and ionic exchange. The spectrum of $Fe_{(I.E.)}$/NC is characterized by at least two resonances, N1 and N2, caused by the $1s\rightarrow\pi^*$ electronic transition for N atoms in NC bonds at 398.8 eV and 401.4 eV, respectively [18,39]. The resonance due to the $1s\rightarrow\sigma^*$ transition is found above 408 eV (N3). Specifically, The R1 resonance at 398.8 eV was assigned to $sp^2$ nitrogen atoms with two carbon neighbors in a pyridine-like configuration, whilst the resonance R3 at 401.4 eV was assigned to three-fold $sp^2$ nitrogen bound to carbon. However, it should be pointed out that it is very difficult to distinguish the N coordinated to the metal from the N uncoordinated. In a previous work, we observed an increase in the full width at half maximum (FWHM) of the resonance and/or a decrease of the intensity due to the interaction with the metal species [18]. In the case of porphyrin, the coordination to the metal center was observed as an increase in the symmetry of the molecules, which was seen as a reduction in the number of resonances appearing in the N1s NEXAFS spectrum. Particularly, the $\pi^*$ region in the N1s NEXAFS spectrum of the porphyrin (398–404 eV) is characterized by three N1s-$\pi^*$ resonances A, B and C at 398.5 eV, 401 eV and 403.9 eV from the N1s orbital of the four chemically equivalent N atoms to 4 unoccupied molecular orbitals formed by the combination of 2p orbitals from the conjugated macrocyclic ligands and the 3d orbitals from the central metal atoms [41].

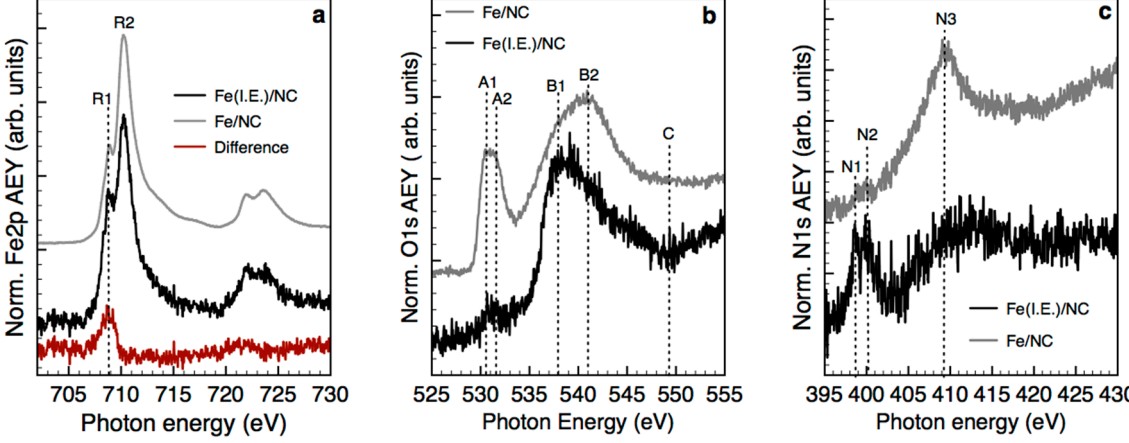

**Figure 8.** Comparisons of the NEXAFS spectra recorded in UHV of the impregnated Fe/NC and the sample obtained via ionic exchange $Fe_{(I.E.)}$/NC: (**a**) the Fe2p AEY spectra and difference spectra; (**b**) the O1s AEY spectra; and (**c**) the N1s AEY spectra.

Thus, the N1s NEXAFS spectrum of Fe-porphyrin does not explain the chemical configuration of N in the impregnated samples. Similarly, the N K edge spectrum reported in literature for the $\varepsilon$-Fe3N nitride phase, an interstitial ordered alloy with Fe atoms in *hcp* array and N atoms occupying the octahedral interstitial sites, is characterized by two resonances in the $\pi^*$ region (397.8 eV and 400 eV) and one resonance in the $\sigma^*$ region at 406.2 eV [42]. This is also very different from the N species present on both the impregnated samples which give arise to a strong and broad $\sigma^*$ resonance with a maximum at about 409.6 eV but a very low intensity in the $\pi^*$ region.

Taking into account the high energy of the broad $\sigma^*$ resonance, we suggest that the nature of the N species from the nitrate precursor is an interstitial N atom in an $sp^3$ bonding configuration, substituting an O atom in the Fe oxyhydroxide phase.

High resolution TEM and electron energy loss spectroscopy (EELS) were used to localize the Fe–N phase. The EELS spectra in Figure 9 clearly depict the presence of a N1s signal collocated with the presence of Fe(III) (Fe2p EELS spectrum) [43]. However, we observed a reduction of the Fe(III) to Fe(II) in certain regions possibly induced by the beam as reported previously [23] for ferrihydrite. Furthermore, whilst we could confirm the presence of N species, the similarity of the acquired spectra

in Figure 9b (N1s spectrum in the inset) with the spectroscopic fingerprint of molecular $N_2$ suggests that the Fe–N bond is broken under the focused electron beam during data acquisition due to the beam sensitivity of the sample.

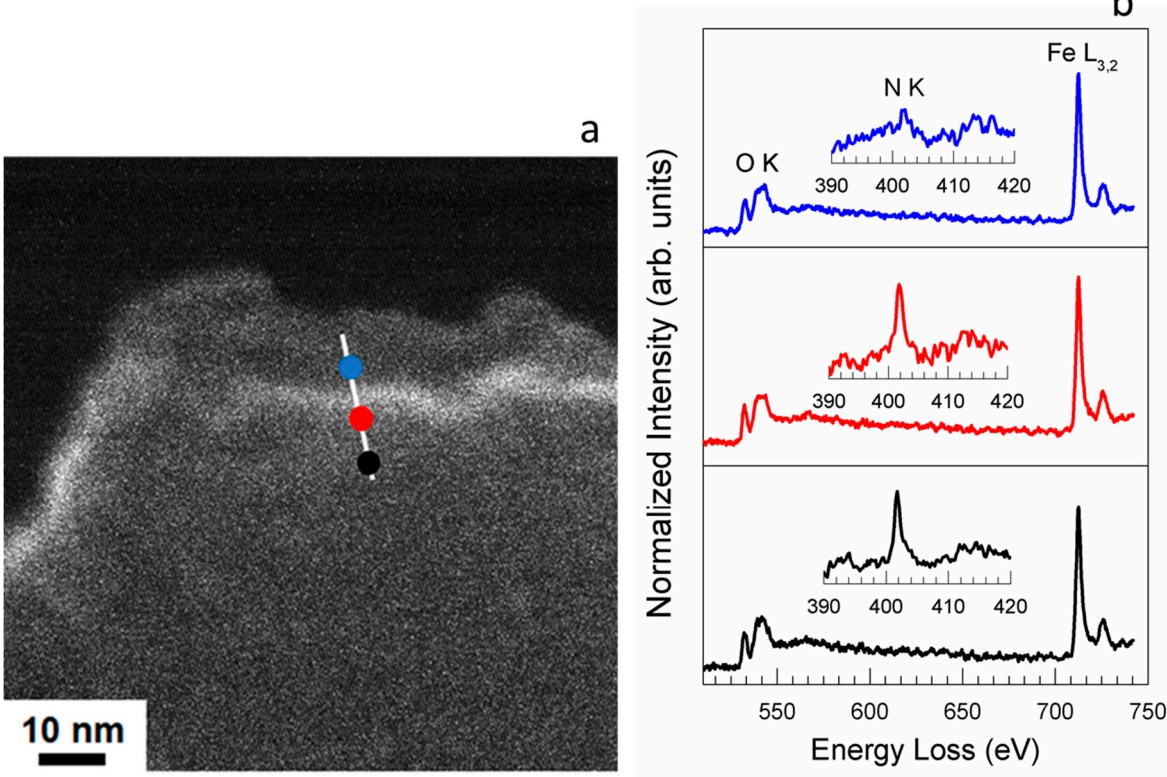

**Figure 9.** The HAADF STEM of the agglomerated ferrihydrite particles (**a**); the Spectrum Image Electron Energy Loss Spectroscopy (EELS) data show the N K edge, O K edge and Fe L3,2 edge region (**b**) from 3 different points as indicated in Figure 9a.

*2.3. Structural Transformation upon Annealing in UHV by X-ray Absorption Spectroscopy, X-ray Photoemission Spectroscopy and TEM*

The Fe/NC sample was thermally annealed in UHV at a temperature above 773 K to understand the thermally induced phase transformation that characterizes these materials. Such information is of general interest not only for the design of specific Fe phases supported on C electro-catalysts but also for understanding the thermal stability in any catalytic applications at mild temperatures and in nonoxidizing environments.

The Fe2p XP spectrum during a temperature ramp in UHV condition is reported in Figure 10a. We can see that, at 473 K, a shoulder appears on the lower BE side and the satellite feature shifts to a lower BE. This observation is consistent with a partial reduction of Fe(III) to Fe(II) [43]. The Fe2p NEXAFS spectrum in Figure 11a shows this more clearly through an apparent change of the intensity ratio of the two main resonances corresponding to a partial reduction of the oxidation state from Fe(III) to Fe(II). The more surface-sensitive O1s XPS spectrum (Figure 10b) shows a reduction of the component assigned to hydroxyl groups ($OH^-$) species more significantly than the $O^{2-}$ species, whereas the O1s NEXAFS spectrum is similar to the one of the fresh samples except for the appearance of the broad feature C at approximately 550 eV, which indicates an increase in the crystallographic order. Increasing further the temperature produces a more significant reduction of the Fe(III) to Fe(II) in the Fe2p NEXAFS spectrum, which now resembles the one reported for wüstite [1,44]. Likewise, the O1s NEXAFS spectrum is characterized by a decrease in the intensities of the resonances A1 and A2 whereas the intensities of the resonances B1 and C increase significantly. Moreover, an additional

resonance in the σ* region appears as a shoulder on the B1 resonance. The spectrum now resembles the one reported in literature for wüstite [37,45].

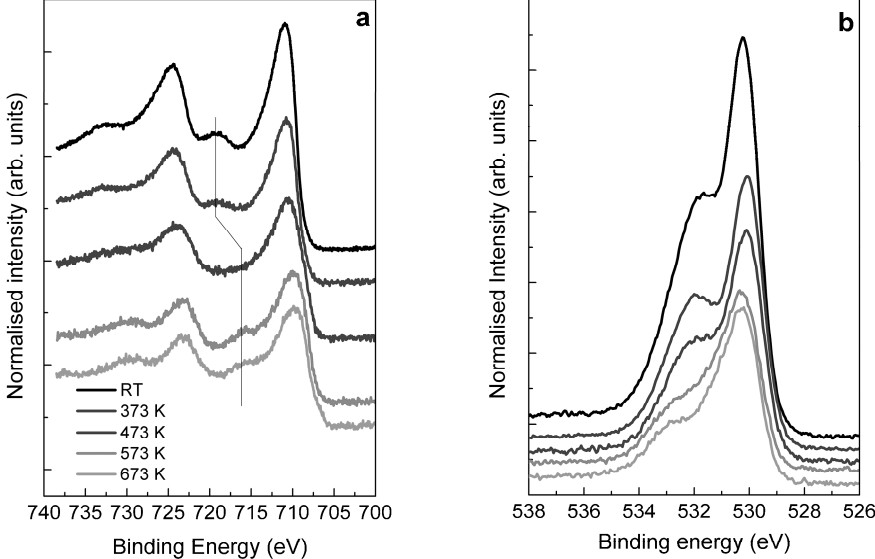

**Figure 10.** The thermal annealing of Fe/NC in UHV: the Fe2p XPS (**a**) and the O1s XPS (**b**).

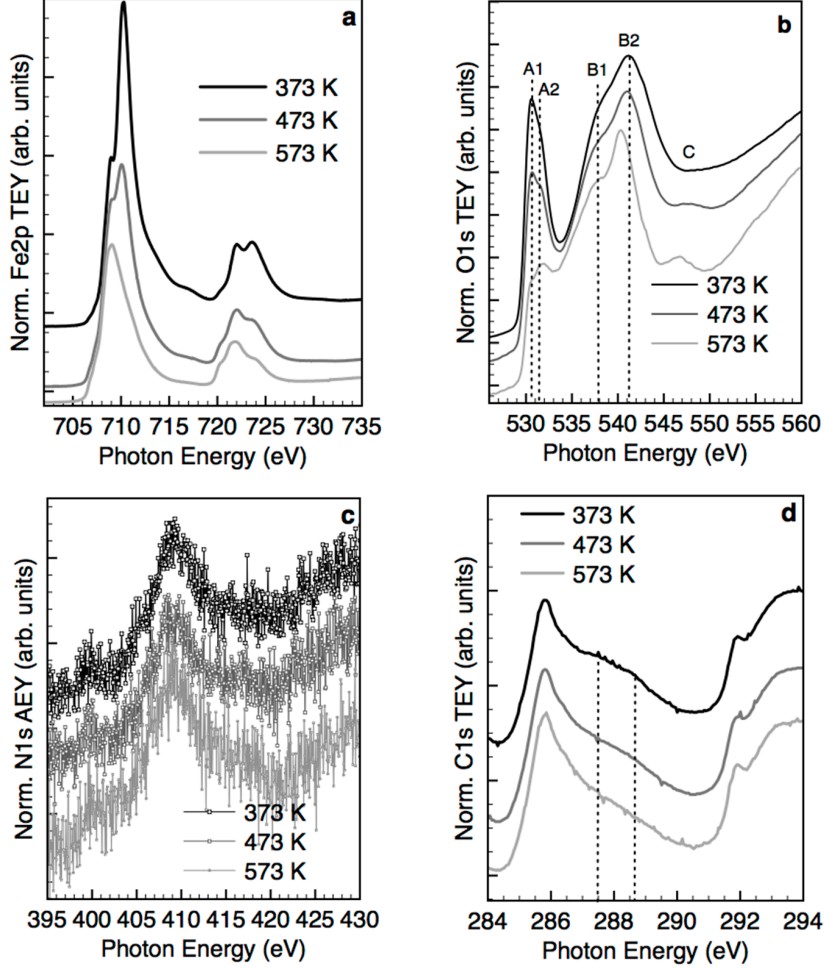

**Figure 11.** The NEXAFS characterization of the impregnated sample Fe/NC in UHV during thermal treatment: (**a**) the Fe2p TEY spectra; (**b**) the O1s TEY spectra; (**c**) the N1s AEY spectra; and (**d**) the C1s TEY spectra.

Interestingly, despite the changes in the O content and the restructuring of the Fe phase, the N1s NEXAFS spectrum does change significantly, whereas slight reductions of the resonances at 287.5 eV and 288.6 eV are observed in the C1s NEXAFS spectrum assigned to C–O species. It can be inferred a good stability of the carbon support and the N-containing Fe phase in nonoxidizing environments.

To understand the structural characteristics giving arise to the Fe2p and O1s absorption spectra in more detail, we followed the structural transformation upon annealing also by in situ HRTEM and selected area electron diffraction (SAED). Figure 12 reports the SAED corresponding to the sample at room temperature (Figure 12a) with the analysis indicating the presence of graphite reflexes. Upon annealing at 473 K, additionally to the graphite reflexes, the ring patterns of ferrihydrite (023) were observed (Figure 12b) [46]. A further increase of the temperature to 573 K leads to the formation of crystalline ferrihydrite domains as seen by the appearance of clear diffraction spots (Figure 11c, green circles). Interestingly, at 573 K, Wüstite is formed as segregated Fe(II) phases coexisting with the Fe(III) ferrihydrite. With increasing the annealing temperature to 773 K, the presence of several phases could be identified. Besides ferrihydrite and wüstite, a mixed Fe(II/III) oxide magnetite is also formed. Particularly, Figure 12d indicates ferrihydrite (013) and (123) reflexes in green, wüstite (002) and (022) reflexes in blue and magnetite (002) in red.

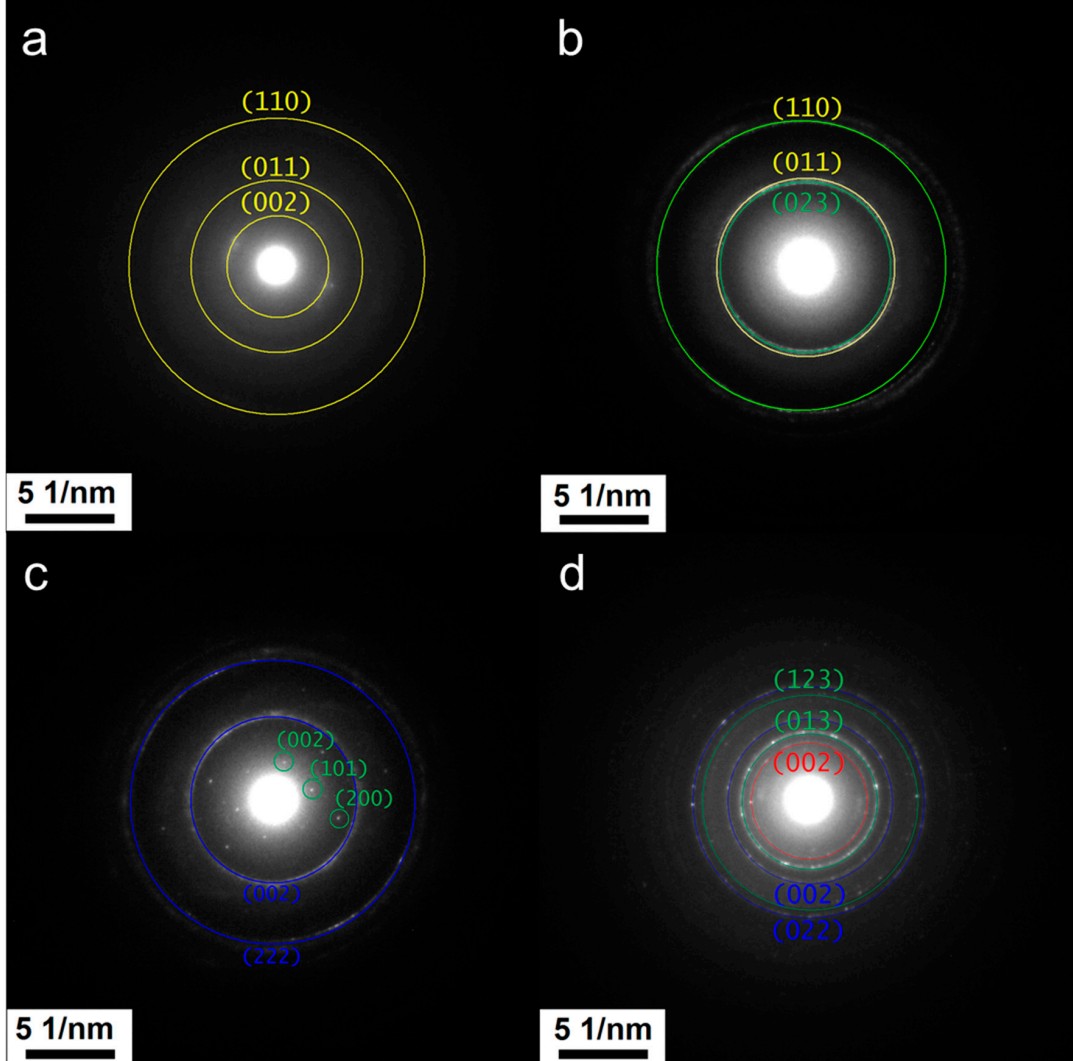

**Figure 12.** The SAED patterns collected during the in situ TEM thermal treatment at RT (**a**), 473 K (**b**), 573 K (**c**) and 773 K (**d**): The assigned structures are labelled by colour (wüstite–blue; graphite–yellow; ferrihydrite–green; and magnetite–red).

This means that, from a *hcp* structure, in which ferrihydrite crystallizes, a rearrangement occurs that transforms the Fe phase in a *fcc* structure. This phase change from hexagonal to cubic phase is apparently assisted also by the presence of C [47]. This result is also interesting from the perspective of a NEXAFS spectra analysis, in which segregated Fe(II) and Fe(III) phases in the same sample would give a similar spectroscopic fingerprint as for magnetite [48], similarly to a building block model [49].

## 3. Discussion

The wet impregnation of a highly ordered functionalized graphitic carbon support with a solution containing Fe species from a ferric nitrate precursor has led to materials with a bimodal distribution of Fe nanostructures: a) highly dispersed single atoms and few atoms clusters and b) large aggregates with ferrihydrite-like structure containing N impurities. This result can be explained rationally, taking into account the surface chemistry of the support, the chemistry of the Fe species in solution and the various interactions between these two phases in each step of the preparation. The chemical functionalization of the carbon surface is carried out with the purpose of inducing defects in the form of chemical species that can anchor metal ions in solution [9,10]. When the carbon support is immersed in an acidic solution containing the metal precursor (for instance, an aqueous solution of

iron nitrate nonahydrate with a density of 6 g·L$^{-1}$ has a pH of 2.3.), the functionalities on the carbon surface as well as the metal ions in solution undergo acid-base equilibria. The electrostatic interactions will induce metal species in solution to interact with the functional groups on the carbon surface of opposite charge [10,13,14].

Concerning the metal species in solution, in analogy to previous studies on the formation of iron oxides in different aqueous media from various ferric salts [50], we expect that their hydrolysis products such as the aquo complex (mono and dimers) and sparingly soluble hydroxides with low nuclearity are formed at this low pH in solution; these polymerize rapidly with increasing pH (from pH > 3) to insoluble products, among which, the oxyhydroxide two-line ferrihydrite phase was reported as the first crystalline compound formed and also the precursor of other Fe (oxyhydr)oxide phases (i.e., akageneite, goethite and hematite). Schwertmann et al. [50] reported also the formation of an ordered Fe(III)-oxyhydroxy-nitrate as a precursor of the six-line ferrihydrite only at pH < 3 because only at this pH is the rate of hydrolysis low enough for this phase to form.

Moreover, Weatherill et al. [24] reported the first evidence for the formation of Fe13 Keggin pre-nucleation clusters during the hydrolysis of a ferric iron solution, which were stable against further aggregation and ferrihydrite NPs formation only at pH < 1. Therein, they also postulate that Keggin clusters can form rapidly within localized areas of high pH (in their experiments at the point of base injection) through the preceding formation of $(Fe(OH)_4)^-$.

Concerning the support surface chemistry, in a previous study, the zeta potential measurements have shown that the surface of carbon nanotubes [15,16] from almost neutral become positively charged only below pH 5 for $NH_3$-functionalized carbon, whereas it is negatively charged for $HNO_3$-functionalized carbon in the overall range of pH. Thus, in these experiments, we expect that the surface chemistry of the carbon support is mostly characterized by protonated N species (pyridinium cations) for NC and by deprotonated O species (carboxylate anions) for OC [15,16].

In the case of the OC, a positive electrostatic interaction between the carboxylates and the mostly abundant mono-hydroxylated and bi-hydroxylated ferric species [17] in solution is realized, and thereafter, specifically chemisorbed ions can act as nucleation centers for the further growth of the nuclei and formation of nanoparticles.

In contrast, this condition is not apparently satisfied in the case of the NC if mono-hydroxylated and bi-hydroxylated complexes were the only species existing in solution. The existence of $(Fe(OH)_4)^-$ species in localized areas of higher pH where the N species are located could explain the dilemmatic capability of N species to coordinate single and low-nuclearity Fe hydroxides clusters in an acidic solution, while the path in which positively charged hexa-aqua cationic complexes are involved would be unfavorable due to the presence of a positive charge on the N functionalities (approx. pKa 8 [15]). It should be noted that amino species are good ligands for ferric cations and were used to prevent the polymerization of these species in solution [17].

Thus, we postulate that, upon the interaction of the carbon support with the Fe-containing solution, Fe monomers and dimers of negative charge for NC (positive charge for OC) rapidly adsorb on the defects of the carbon surface, forming highly dispersed Fe clusters and single Fe atoms. For a highly graphitic C support, like the one we are using in this study, the number of functional groups that can be introduced by these chemical posttreatments ($NH_3$ or $HNO_3$) is limited to the availability of edge sides and vacancies. Therefore, it is normally low, and as a consequence, its ionic exchange capacity is limited. Comparatively, single atoms or few atoms clusters are relatively more abundant on the sample containing the N-functionalized carbon support due to the high thermal stability of these functionalities upon the subsequent thermal treatment (compared to the carboxylates which decompose already at 473 K). The electronic structure fingerprint of this minority of species was identified here using Fe2p photoelectron and absorption spectroscopies, and we found that these are Fe(II) species under UHV condition.

However, the majority of the Fe species are present as relatively large particles, showing high morphological anisotropy and particles size inhomogeneity but the same crystallographic phase

regardless of the carbon support used. The bulk structure determined by means of X-ray absorption fine structure spectroscopy was consistent with the ferrihydrite structure; however, TEM showed that the nanostructured particles are poorly ordered. It is, therefore, more correct to describe this phase as a ferrihydrite-like Fe(III)-oxyhydroxide which contains also thermally stable Fe–N species.

This agglomerated ferrihydrite-like phase is formed due to the higher local pH in proximity of the C surface, which favor its precipitation (at pH > 3). Furthermore, during the drying process overnight, we postulate that insoluble Fe(III)-oxyhydroxy-nitrate species are formed via a slow kinetic process, which aggregate into nanoparticles and deposit on the C surface with no specific interaction with the C support.

In the subsequent step, the sample is subjected to a thermal annealing in $N_2$ at 523 K for 4 h, which finally gives the ferrihydrite-like/C material under study. The thermal decomposition mechanism of Fe(III) nitrate under $N_2$ was previously studied by using infrared spectroscopy, thermo-gravimetric analysis and X-ray diffraction (XRD) [50]. Therein, it was shown that the denitrification of Fe(III) nitrate was completed at 413 K, and above this temperature, only the evolution of water was observed whilst a small number of hydroxyl groups remaining on the surface were identified [51]. Moreover, in the same work, XRD indicated that hematite was formed from thermal decomposition of Fe(III) nitrate in inert atmosphere only at a temperature above 573 K.

This is in contrast with our findings showing that N is retained in the structure with an $sp^3$ electronic configuration and remains invariant up to 773 K. We suggest that, during the final thermal annealing of the dried sample precursor at 523 K, the interaction of the Fe(III)-oxyhydroxy-nitrate with the carbon support might stabilize N atoms in the ferrihydrite structure at the interface with the C support.

We also characterized the thermally induced transformations of a freshly prepared ferrihydrite/C sample from room temperature to 773 K, which showed a series of unexpected phase transformations. Firstly, it was noticed that the thermal annealing up to 473 K induced the ordering of the ferrihydrite phase. This means that the exposure of the sample to moisture/air after the final synthesis step (thermal decomposition of the precursor Fe(III)-oxyhydroxy-nitrate at 523 K) produces structural modifications (i.e., surface hydration). Secondly, the thermal annealing of the ferrihydrite/C sample at 573 K led to the formation of a segregated phase of wüstite (Fe(II)O) coexisting with the ferrihydrite phase. Only at 773 K, the formation of a mixed oxide Fe(II, III) magnetite was observed. Similarly to Ferrihydrite in biological systems [1,47], we suggest that the C support has a strong influence in favoring the reduction of Fe(III)ferrihydrite to Fe(II) wüstite rather than the condensation of OH to form Fe(III) hematite.

This work clarifies the synthesis conditions for Fe/C materials for achieving a homogeneous immobilization of the active species on the graphitic support with a similar size and same chemical nature when using wet impregnation. Particularly important is the choice of the metal loading in relation to the nature and abundance of the functionalities on the carbon surface. The use of a low-concentration ferric solution and a highly N-functionalized carbon support enables the control of nucleation and growth of the Fe species by the interfacial acid/base equilibria realized in the first step of the synthesis, which guarantees a high metal dispersion. With ferric solutions too concentrated, the abundance of N species on the carbon support is not sufficient to quantitatively immobilize the Fe species in solution in the first step, and as a consequence, larger particles are formed in the subsequent drying and thermal annealing steps. In the case of O-functionalized carbon supports, the low thermal stability of some of the oxygen species leads the initially dispersed clusters to agglomerate in the third step of the synthesis, and therefore, in the case of OC, other synthetic routes could me more appropriate to attain a high dispersion.

In a previous work, these materials were tested for the electrochemical $CO_2$ reduction reaction [6]. These large agglomerates were found not only to be inactive in the potential range for selective $CO_2$ reduction but also detrimental for the catalytic activity as they block some of the active N–Fe ensembles formed at the interface between the few atoms Fe(II/III)-clusters and the N-functionalized graphene

edges of NC. Generally, large metal/metal oxide NPs weakly interacting with the carbon support are mobile under reaction conditions and undergo agglomeration and coalescence [7,18–20]. This is very often accompanied by the worsening of the catalytic performances with reaction time, which is explained in terms of a reduced exposed active surface or changes in the electronic structure of the exposed surface. For application-oriented materials design, whilst the nature of the desired active phase to be stabilized on the support is not always known a priori, attaining a high homogeneity in the particles size after the final thermal treatment is indicative of a high stability of the supported phase through sufficiently strong metal support interactions. In fact, a successful catalyst design for a specific application would requires active phase/support interactions strong enough to resist the reaction conditions in which it is used.

## 4. Materials and Methods

### 4.1. Synthesis of Impregnated Fe/OC, Fe/NC and Fe$_{(I.E.)}$/NC

We used a TorayTM Carbon paper TGP-H-030 (FuelCellStore.com) with a thickness of 0.1 mm and dimensions of approx. $0.8 \times 0.8$ cm$^2$ (approx. 4.2 mg) as a support for the metal oxyhydroxide phase. Prior to the impregnation of the metal precursor, the carbon cloth was heated to 393 K in HNO$_3$ (250 mL, 70% Sigma-Aldrich, Dorset, UK) for 4 h, followed by drying in static air overnight at 373 K. This treatment allows the introduction of oxygenated functionalities. This sample is referred to as OC. In a second step, the HNO$_3$-treated samples were put in a tube furnace under 50 mL min$^{-1}$ NH$_3$ (99.98% Ammonia Micrographic, BOC Linde, Munich, Germany) at 873 K for 4 h. Afterwards, the samples were cooled down to 323 K in NH$_3$ and further to room temperature in N$_2$ (50 mL min$^{-1}$, BOC Linde, Munich, Germany). This sample is referred to as NC. Both OC and NC were used to immobilize Fe. The Fe containing samples (Fe wt% = 1) were obtained via the incipient wetness impregnation of an Fe(NO$_3$)$_3$•9H$_2$O solution in H$_2$O/ethanol (24:1). An aliquot of 100 μL of a 3 g/L solution was used. The solution was added dropwise to the single carbon cloth piece, paying attention that the wetting of the carbon paper piece was homogeneous. The impregnated pieces of carbon paper were dried at room temperature in air overnight. Afterwards, the samples were heated at 523 K in N$_2$ (50 mL min$^{-1}$, BOC Linde, Munich, Germany) for 3 h in order to achieve the decomposition of the metal precursor. The samples were cooled down to room temperature in N$_2$ and afterwards exposed to air/moisture, prior to characterization.

The sample Fe$_{(I.E.)}$/NC was obtained via the ionic exchange route. Accordingly, the functionalized NC was placed in 25 mL of a FeCl$_3$ solution (0.75 g/L). Subsequently, 62.5 μL of concentrated HCl (37% from Sigma-Aldrich, Dorset, UK) were added to this solution, and the solution was sonicated for 5 min. Afterwards, the solution was heated up to 333 K for 1 h using a steam bath. The sample was then dried overnight in air and characterized.

### 4.2. X-ray Photoelectron and Absorption Spectroscopy

X-ray photoelectron spectroscopy (XPS) and near-edge X-ray absorption fine structure (NEXAFS) measurements in the soft X-ray regime were carried out at the ISISS end station and beamline at Helmholtz-Zentrum Berlin (HZB) (Berlin, Germany). The freshly prepared samples from an atmospheric environment were directly exposed to a vacuum ($10^{-7}$ mbar) in the XPS chamber. The XPS measurements were performed by applying a suitable excitation energy corresponding to a kinetic energy (KE) of the photo-emitted electrons of 450 eV and/or 150 eV for the core levels Fe2p, C1s, O1s and N1s. The energy pass Ep was normally set to 20 eV. The core levels envelopes were fitted using the Casa XPS software after subtraction of a Shirley background.

The fittings of the Fe2p, O1s and N1s were performed by considering as many components with a Gaussian–Lorentzian (GL) line-shape as needed to describe consistently structural changes among the samples and upon temperature programmed treatment. The fitting of the spectra was done by constraining the peak position by ±0.05 eV. The peak area ratio between the Fe2p3/2 and Fe2p1/2

spin-orbit split transitions was constrained approximately to the theoretical value of 2:1, and the distance between the two-spin orbit split transition was 13.5 eV. The binding energies (BEs) were referenced to the maximum of the C1s core level peak (284.3 eV) measured after every other core level measurement at the corresponding excitation energy. The quantification of the elemental composition was carried out according to a homogeneous model distribution. For quantification, the spectra have been normalized to the impinging photon flux.

The Auger Electron Yield (AEY) NEXAFS spectra were recorded with an analyzer setting of 50 eV pass energy (Ep) and electron kinetic energies (KE) of 700 eV, 520 eV, 350 eV and 240 eV for Fe L, OK, NK and CK, respectively. The beam-line setting was exit slit (ES) 111μm and fix focus constant (cff) 1.4 (cff = cosα/cosβ). The kinetic energy window was chosen such as to avoid photoelectrons moving through the NEXAFS spectrum while sweeping the excitation energy, while a broad Ep was necessary to obtain reasonable intensity. The exit slit value chosen enables an optimal compromise between a high photon intensity and a good spectral resolution. The higher order suppression operation mode of the monochromator was applied (fix focus constant cff = 1.4) to avoid contributions to the background in the NEXAFS spectra that might complicate the intensity normalization of the spectra on impinging photon flux. The sample heating was assured by an IR-laser mounted on the rear part of the sample holder. Temperature control was realized using two K-type thermocouples.

### 4.3. Electron Microscopy Techniques

The bright field (BF) and high angle annular dark field scanning transmission electron microscopy (HAADF STEM) images were acquired on a probe corrected ARM200F at the ePSIC facility (Diamond Light Source, Didcot, UK) equipped with a cold-FEG and operated at an acceleration voltage of 200 keV, enabling a resolution of 0.78Å. The measurement conditions were a CL aperture of 30 μm, convergence semiangle of 24.3 mrad, beam current of 12 pA, and scattering angles of 0–10 and 35–110 mrad for BF and HAADF STEM respectively. The SEM analysis was performed on a Zeiss Ultra SEM operating at an acceleration voltage of 1.6 and 20 keV. The samples were grinded between two glass slides and then deposited onto 3 mm holey carbon Cu-TEM grids and SiNx chips for ex situ and in situ analyses, respectively.

The in situ TEM experiments were carried out utilizing a DENSSolutions wildfire in situ holder. For an accurate magnification calibration, the diffraction pattern from Au nanoparticles were acquired under identical experimental conditions

### 4.4. Raman

The Raman measurements were carried out on a Renishaw InVia Raman microscope ($\lambda$ = 473 nm) with a 50× objective.

## 5. Conclusions

In this work, we report a detailed structural characterization of Fe-impregnated C catalysts using a multi-technique approach. This includes synchrotron-based high-energy resolution X-ray spectroscopy and atomic-resolution transmission electron microscopy. We identify two different mechanisms of Fe immobilization on functionalized graphitic carbon as a cause of the high morphological anisotropy, broad particles size distribution and complex Fe speciation:

(a)　Single atoms and clusters are formed during the first impregnation step from the chemisorption of soluble Fe(III) species on the heteroatoms at the graphite edges. In the specific case of N-functionalized graphitic carbon, the high local pH in proximity to the functional groups induces clustering of the Fe species in parallel to their chemisorption on the carbon surface. Specifically, anionic Fe(III) species interact with the pyridine N functional groups on the C surface (or few basic and thermally stable O species on the OC support) and are stabilized in an Fe(II)

oxidation state. Moreover, more abundant cationic Fe(III) species interact with carboxylates species on the OC surface.

(b)    The majority of the Fe(III) species in solution slowly polymerize during the drying process, forming Fe(III)-oxyhydroxy-nitrate NPs, which deposit on the carbon surface with no specific interaction with it. The thermal annealing in $N_2$ transforms these Fe(III)-oxyhydroxy-nitrate NPs into big agglomerates of ferrihydrite NPs containing N impurities as Fe–N–O bonds in $sp^3$ configuration but with a different spectroscopic fingerprint than nitrate moieties.

For application-oriented materials design, attaining a high homogeneity in particles size is indicative of the occurrence of specific metal support interactions, a prerequisite for high catalyst stability under reaction conditions. This work shows that, in order to achieve this goal, the synthesis of Fe/C catalysts via impregnation must be designed in such a way that, during the first step, the Fe species in solution are quantitatively immobilized on the carbon support through the occurring interfacial acid/base equilibria. To this purpose, particularly important is the metal loading which must be chosen in relation to the nature and abundance of the functionalities available on the carbon surface for metal coordination.

**Author Contributions:** R.A. and M.E.S. conceived and designed the experiments; R.A. performed the XPS and NEXAFS experiments and analyzed the data; M.E.S. performed the TEM experiments and analyzed the data.

**Acknowledgments:** The authors would like to thank Panayiotis Tsaousis and Verena Streibel for their support during beamtime and Tina Geraki (DLS) for the assistance with measuring the Raman spectra. We thank HZB for the allocation of synchrotron radiation beamtime for the proposal ST 15202970. This project has received funding from the European Union's Horizon 2020 research and innovation programme under grant agreement No 730872.

**Conflicts of Interest:** The authors declare no conflict of interest.

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
