# Peer review of "On the High Structural Heterogeneity of Fe-Impregnated Graphitic-Carbon Catalysts from Fe Nitrate Precursor"

_catalysts, doi:10.3390/catal9040303_

Round 1

Reviewer 1 Report

The manuscript has been prepared with good information.  Samples and data have been used in a properly way. Therefore, I would like to mention some points about the aforementioned paper in order to be taken into account by the authors. For these reasons I recommend the publication of this manuscript with a major revision.

Please consider following general and specific recommendations that are strictly required to improve the quality of the manuscript:

1. The title might be changed from “On the high structural heterogeneity of Fe impregnated carbon catalysts from Fe nitrate precursor” to “On the high structural heterogeneity of graphite-supported Fe impregnated carbon catalysts from Fe nitrate precursor” so as to be more accurate with the point of these study.

2. In “1. Introduction” lines 27-29, the authors mention that “Fe oxides and oxyhydroxides exist in nature in a large variety of structures and play a primary role in many natural geological and biological processes.” Recent developments include activated Carbon Modified by Iron and Manganese Oxides, with success, so they could make a reference in order to give the importance and specification of the use of Fe-modified materials (the authors can consult the paper Gallios et al., 2017: “Adsorption of Arsenate by Nano Scaled Activated Carbon Modified by Iron and Manganese Oxides, 2017. Sustainability 9(10) DOI 10.3390/su9101684.

3. In “4. Materials and Methods 4.1. Synthesis of impregnated Fe/OC, Fe/NC and Fe i.e./NC, please give in a Table the aforementioned synthesized materials, with their characteristics.

4. For the material “Fe(i.e.)/NC”, I would prefer a different name because the acronym (i.e.) standing for ionic exchange synthetic route, it is not very clear as mainly refers to an example of Fe and not ionic exchange. I suggest the authors to replace it across the manuscript with the term “Fe(I.E.)/NC”. In addition I cannot understand clearly, the significance of these material.

5. The Abstract and the conclusions are poor, please try to meliorate them.

Best regards

Author Response

We thank the referee for reviewing our manuscript and providing useful recommendations to improve our manuscript. The manuscript has been revised following the points raised. An explanation of how we met the referee´s recommendation follows:

1) In order to avoid repetion of words (C and graphite in the version proposed by the referee) while meeting the referee good point we propose the following title: "On the high structural heterogeneity of Fe impregnated graphitic-carbon catalysts from Fe nitrate precursor".

2) Indeed we are committed to provide a manuscript as comprehensive as possible on the application of Fe/C systems, so we took on board the suggestion from the referee and cited the mentioned article in this version.

3) A summary of the samples characteristics is reported in table 1. We believe that these are the meaninful numerical characteristic of the samples which can be included in a table.

4) The notation i.e. was changed in I.E. as suggested by the referee in both the text and the figures. Additionally at lines 94-97 page 3, the following sentencewas added.

An additional sample was prepared via ionic exchange from iron chloride aqueous solution (referred to as Fe(I.E.)/NC) as a reference for the spectroscopic analysis of Fe, N, and O species in the impregnated samples.

This clarify the point of preparing this sample, which was purely necessary to discriminate from a spectroscopic viewpoint the N species on the C support from those originated from the nitrate precursor. This concept was several times clarified in the manuscript.

5) We significantly improved the abstract and the conclusions sections

Reviewer 2 Report

In my opinion, it is a very  nice and well written paper, that can be accepted as it is.

Author Response

We are very grateful to the referee for reviewing our manuscript and recommending its publication as it is.

Reviewer 3 Report

This study focused on the synthesis and advanced characterization (tools) of C-supported Fe catalysts. The major findings of this work include the discovery of a thermally-induced phase transformation of oxy-hydroxide from precursor into several oxide phases. It was also found that there existed a thermally-stable N-containing impurities in the structure. The interactions with the C support and the structural dynamics induced by the thermal treatment were thought to rationalize the heterogeneity observed in these catalysts. Indeed, the results of this work could have implications on how to control synthesis of C-supported Fe catalysts. I strongly recommend this manuscript be accepted with very minor revision. For example, authors should include the conditions under which homogeneous distribution of Fe species on the support be attained. In addition, are these same materials been tested, and if yes, what catalytic applications? What are the potential consequences of large heterogeneities on the activities of the catalysts?

Author Response

We are very grateful to the referee for providing this feedback. We indeed took on board the referee´s suggestions and made some additional explanation of this work and linked it to our previous work.

Indeed these materials were previously used for the CO2RR and a mention of that work, which substantiate and motivated this work, is now included in this manuscript version. In addition, we discuss why it is important to obtain a homogeneous distribution of partiles size and phases.

These considerations are included at page 17-18 lines 494-520  and page 20 lines 615-623.

The text is also copied below:

In the discussion part: This work clarifies the synthesis conditions for achieving a homogeneous immobilization of active species on the graphitic support with similar size and the same chemical nature when using wet impregnation for the synthesis of Fe/C materials.  Particularly important is the choice of the metal loading in relation to the nature and abundance of the functionalities on the carbon surface. The use of low concentration ferric solution and highly N-functionalized carbon supports enables that the  nucleation and growth of the Fe species is controlled by the interfacial acid/base equilibria realized in the first step of the synthesis, which guarantee a high metal dispersion. With ferric solutions too concentrated, the abundance of N species on the carbon support is not sufficient to quantitatively immobilize the Fe species in solution in the first step and as a consequence larger particles are formed in the subsequent drying and thermal annealing steps. In the case of O-functionalized carbon supports, the low thermal stability of some of the oxygen species leads the initially dispersed clusters to agglomerate in the third step of the synthesis and therefore in the case of OC other synthetic routes could me more appropriate to attain high dispersion.

In a previous work, these materials were tested for the electrochemical CO2 reduction reaction [3]. These large agglomerates were found not only to be inactive in the potential range for selective CO2 reduction but also detrimental for the catalytic activity as they block some of the active N-Fe ensembles formed at the interface between the few atoms Fe(II/III)-clusters and the N-functionalized graphene edges of NC. Generally, large metal/metal oxide NPs weakly interacting with the carbon support are mobile under reaction conditions and undergo agglomeration and coalescence [4, 13]. This is very often accompanied by the worsening of the catalytic performances with reaction time, which is explained in terms of reduced exposed active surface or changes in the electronic structure of the exposed surface. For application-oriented materials design, whilst the nature of the desired active phase to be stabilized on the support is not always known a priori, attaining high homogeneity in particles size after the final thermal treatment is indicative of a high stability of the supported phase through sufficiently strong metal support interactions. In fact, a  successful catalyst design for a specific application would requires active phase/support interactions strong enough to resist the reaction conditions in which it is used.

Conclusions:

For application-oriented  materials design,  attaining high homogeneity in particles size is indicative of the occurrence of specific metal support interactions, a prerequisite for higher catalyst stability under reaction conditions. This work shows that in order to achieve this goal, the synthesis of Fe/C catalysts via impregnation must be designed in such a way that during the first step the Fe species in solution are quantitatively immobilized on the carbon support through the occurring interfacial acid/base equilibria. To this purpose, particularly important is the metal loading which must be chosen in relation to the nature and abundance of the functionalities available on the carbon surface for metal coordination.

Round 2

Reviewer 1 Report

The authors took my comments into concideration and now the paper,  according to my opinion,  is ready for publication